# Estimation and Analysis of GNSS Differential Code Biases (DCBs) Using a Multi-Spacing Software Receiver

**DOI:** 10.3390/s21020443

**Published:** 2021-01-10

**Authors:** Ye Wang, Lin Zhao, Yang Gao

**Affiliations:** 1College of Intelligent System Science and Engineering, Harbin Engineering University, Harbin 150001, China; wangye123@hrbeu.edu.cn; 2Department of Geomatics Engineering, University of Calgary, Calagry, AB T2N 1N4, Canada; ygao@ucalgary.ca

**Keywords:** differential code biases (DCBs), correlator spacing, multi-spacing DCB (MSDCB), GNSS

## Abstract

In the use of global navigation satellite systems (GNSS) to monitor ionosphere variations by estimating total electron content (TEC), differential code biases (DCBs) in GNSS measurements are a primary source of errors. Satellite DCBs are currently estimated and broadcast to users by International GNSS Service (IGS) using a network of GNSS hardware receivers which are inside structure fixed. We propose an approach for satellite DCB estimation using a multi-spacing GNSS software receiver to analyze the influence of the correlator spacing on satellite DCB estimates and estimate satellite DCBs based on different correlator spacing observations from the software receiver. This software receiver-based approach is called multi-spacing DCB (MSDCB) estimation. In the software receiver approach, GNSS observations with different correlator spacings from intermediate frequency datasets can be generated. Since each correlator spacing allows the software receiver to output observations like a local GNSS receiver station, GNSS observations from different correlator spacings constitute a network of GNSS receivers, which makes it possible to use a single software receiver to estimate satellite DCBs. By comparing the MSDCBs to the IGS DCB products, the results show that the proposed correlator spacing flexible software receiver is able to predict satellite DCBs with increased flexibility and cost-effectiveness than the current hardware receiver-based DCB estimation approach.

## 1. Introduction

Differential code biases (DCBs) in global navigation satellite systems (GNSS) are the inter-delay differences between two or three frequencies of GNSS signals. DCBs occur because of hardware imperfections inside receivers and satellites, and are contained in GNSS receiver observations [1]. However, DCBs must be known for GNSS applications, such as total electron content (TEC) determination, which derives TEC from receiver observations [2,3,4]. DCBs can cause errors of several meters in TEC estimates if they are ignored and can even result in negative ionospheric delay values [5]. DCBs are usually considered to consist of two components: satellite-related DCBs and receiver-related DCBs [6]. Two different categories of methods are available for determining DCBs based on data from ground GNSS receiver stations. The first approach is to determine satellite and receiver DCBs simultaneously through local or global ionospheric TEC modeling [7,8,9]. Therefore, the errors in ionospheric model coefficients will lead to errors in the satellite and receiver DCB estimates using this method. The second approach is to predict receiver and satellite DCBs through GNSS code measurement differences after the ionospheric delays have been modeled. The accuracy of the DCBs estimated by this method is determined by the accuracy of the ionospheric products [10,11,12].

Some new methods have been proposed to estimate GNSS DCBs. For instance, a new mapping function that exploits the ionospheric varying height (IVH) has been developed to increase the performance of DCB estimates [13]. The onboard receiver of the China Fengyun-3C agile satellite has been used to estimate GNSS DCBs, which indicates that the onboard receiver DCB is very stable [14]. Regular receivers on earth are not as stable as an onboard receiver. In [15], DCBs of a new BDS satellite (BDS-3) are estimated based on an undifferenced and uncombined model of triple-frequency observations. To date, these DCB estimation methods are all based on long-period multi-station or single-station observations (at least 24 h) from structural fixed hardware receivers, including their signal tracking loops. However, the DCBs estimated by these methods are at delayed by least one day, and cannot be used for real-time measurements. The characteristics of DCB over a short time period (e.g., 15 min) have not been discussed, and the in-receiver signal processing process has also not been considered. These are the focuses of our research.

Signal correlation inside a receiver is a process that compares various time signals to the local signal and determines whether they have anything in common. The correlator in GNSS receivers is used to align the replica code with the transmitted code from GNSS satellites. The correlator spacing can determine how and where the two signals are correlated, which induces biases in GNSS observations [15]. GNSS signals usually have chip shape distortion when the GNSS signal is transmitting and receiving in the radio frequency (RF) filter of the satellite and receiver front-end. These chip shape distortions will lead to deviations from the ideal triangular shape of the receiver’s correlation function. These deviations will cause a phase shift in the code tracking point and lead to a pseudorange bias in the code observation [16,17]. In [18], a “second-order step” model was developed as a combination of analog distortion and digital distortion effects. Analog distortions manifest as ‘‘ringing’’ after the chip transitions [19,20]. In [21], Hauschild and Montenbruck characterized the correlator spacing dependency of the code biases of GNSS signals using a particular firmware. This firmware can be used to collocate with different early-late (E–L) correlator spacings of each tracking channel to track the signal of one satellite at the same time. Furthermore, inconsistent code measurement biases of a mixed receiver type reference station network will lead to inconsistent influences on the DCB estimation process, which should be investigated. How the in-receiver correlation process affects DCB estimation must be analyzed.

This study proposes a new approach for GNSS satellite DCB estimation based on a correlator spacing flexible software receiver and is called the multi-spacing DCB (MSDCB) estimation method. This method aims to improve the real-time performance of satellite DCB estimation and make use of the correlator spacing-induced biases in the DCB estimation approach. The main idea of the MSDCB method is to increase the observation number of a single station with different correlator spacings since the single-station DCB estimation method can also offer accurate satellite DCBs. In this paper, the influence of the correlator spacing on GNSS pseudorange observations is first discussed from a theoretical aspect. Then, the satellite DCB estimation method is described based on International GNSS Service (IGS) GNSS stations and a multi-spacing software receiver [22,23]. Next, the influence of the correlator spacing on multi-station DCB estimation is investigated following a stability analysis of the DCB estimation over a short time period (15 min). A set of DCB products are determined using the proposed MSDCB approach and evaluated by comparing them to those estimated by the IGS Center for Orbit Determination in Europe (CODE) group [24]. Finally, a summary of the work and future research suggestions are presented.

## 2. Methodology

To analysis and predict satellite DCBs with observations of different the correlator spacing, the correlator spacing influence on pseudorange measurements and the DCB estimation methodology along with MSDCB are introduced in this section. We start with the discussion of correlator spacing influence on the delay locked loop (DLL) based code tracking loop and code measurements [15,22]. The DCB estimation method is elaborated with an introduction to the observation of preprocessing technology.

### 2.1. Correlator Spacing Influence on Pseudorange Measurements

GNSS signals usually have chip shape distortion when the GNSS signal is transmitting and receiving in the radio frequency (RF) filter of the satellite and receiver front-end chain, respectively. The GNSS code signal chip shape is commonly considered as a perfect rectangular shape under digital signal processing theory [21]. There is an imbalance of the duration between the high and low bits in one cycle of the digital chip due to the digital distortion of the high bit falling edges compared to the rising edge. These distortions will lead to deviations from the ideal triangular shape of the receiver’s correlation function. These deviations will cause a phase to migrate in the code tracking point and lead to a bias in the code observation.

A number of code tracking methods are used to track the range code inside GNSS receivers, and DLL, which is used in this paper, is the most widely used code tracking method [22,25,26,27]. The DLL code tracking loop is used to duplicate the range code (GPS L1 C/A) in the intermediate frequency (IF) signal to the perfectly synchronized local range code. The DLL uses the early and late correlator outputs to calculate the synchronization error between the IF signal range code and the local replica range code via a code discriminator. The correlation process inside the correlators is a predetection integration process [15]. The code discriminator is used to calculate the signal parameter error information of the correlator outputs between the early and late branches [22]. The loop filter is used to reduce signal noise to produce an accurate estimate of the input signal. The filtered error is used as the input of the local code number-controlled oscillator (NCO), which is the controller of the local range code, to modulate the shift register while generating the local range code. The structure of the code tracking model under the time domains shown in Figure 1 [15,22], r[t;τ(t)] is the DLL input signal (IF), which involves the receiver antenna bias, receiver front-end bias, satellite biases, and atmosphere delays. $\hat{r}\left[t;\hat{\tau }\left(t\right)\right]$ is the replica range code generated by the local replica generator, which contains the receiver hardware biases. τ is the real code phase of IF signal, and δτ(t) is the code phase error computed by the discriminator. $\dot{\tau }\left(t\right)$ denotes the filtered code phase error, and $\hat{\tau }\left(t\right)$ is the input of local replica generator.

The discriminator output δτ(t) can be expressed as the local replica code delay estimate error ${\hat{\tau }}_{\epsilon }$, it can be modeled as:(1)$${\hat{\tau }}_{\epsilon }=\tau^{s}+\delta \tau^{i,s}+\delta \tau^{i,c}+\delta \tau^{i,d}+\delta \tau^{i,t}+\delta \tau_{j,c}+\delta \tau_{j,s}+\delta \tau_{j,t}$$
where $\tau^{s}$ is the ideal satellite code phase, $\delta \tau^{i,s}$ is the satellite clock offset due to satellite oscillator phase jitter, $\delta \tau^{i,c}$ is the satellite chip shape distortion generated by satellite oscillator phase jitter, and $\delta \tau^{i,d}$ and $\delta \tau^{i,t}$ are the propagation -induced code delay and satellite thermal error, respectively. $\delta \tau^{i,s}$, $\delta \tau^{i,c}$, $\delta \tau^{i,d}$ and $\delta \tau^{i,t}$ are satellite-related pseudorange errors. The other three: $\delta \tau_{j,c}$, $\delta \tau_{j,s}$, and $\delta \tau_{j,t}$ are receiver-related pseudorange errors: $\delta \tau_{j,c}$, and $\delta \tau_{j,s}$ are the receiver oscillator offset and the receiver chip shape distortions-induced pseudorange errors, respectively. $\delta \tau_{j,t}$ is the receiver thermal error.

In addition, the correlator spacing is an important parameter that affects ${\hat{\tau }}_{\epsilon }$ due to the approximation of the early and late correlator nonconvexity cost functions. This error will be transmitted to the code observations and exhibited as pseudorange biases. In Equation (1), the error sources of the local replica code delay estimate error are errors of code measurements, and satellite and receiver chip shape distortions can induce unignored pseudorange biases. The undifferenced pseudorange (*P*) of receiver *j* for satellite *i* can be modeled as:(2)$$P_{j}^{i}=\rho_{j}^{i}+c\left(\tau^{i}-\tau_{j}\right)+I^{i}+M\tau +d^{i}+d_{j}+\epsilon_{j,c}^{i}$$
where $\rho_{j}^{i}$ is the absolute range, and $\tau_{j}$ and $\tau^{i}$ are the receiver and satellite clock errors, respectively. *c* is the speed of light in vacuum, and $I^{i}$ and τ are the ionosphere and the tropospheric delays, respectively. These delays are transformed from propagation delay ($\delta \tau^{i,d}$), *M* is the mapping function of the zenith tropospheric delay (ZTD) τ. $d^{i}$ and $d_{j}$ are the satellite hardware-induced code bias and receiver hardware-induced code bias, respectively, and $\epsilon_{j,c}^{i}$ is the satellite and receiver code thermal noise. References [21,28] proved that the carrier phase and pseudorange observations measured by different correlator spacings are independent and can be used to classify and estimate satellite biases.

### 2.2. GNSS Observation and Pre-Processing

GNSS dual-frequency observations are commonly employed to estimate TEC for monitoring variations in Earth’s ionosphere. The inter-frequency delay difference between the GPS L1 and L2 signals must be considered during the TEC estimation process [2]. DCBs are one of the main parts of TEC estimation. To estimate precise DCBs, the differential code observations must be smoothed before using the least squares (LS) method. The smoothing process avoids the estimation of carrier geometry-free, nonreal, ambiguities with the carrier-to-code levelling method [5,29].

GNSS observations are used to calculate the ionospheric and DCB measurements [2,5,30,31]. The ionospheric TEC and DCB measurements are calculated through several steps as described below. The dual-frequency GPS observations can be shown as Equation (3) from Equation (2) for the GPS L1 and L2 signals:(3)$$\left\{\begin{array}{c} P_{1}=\rho +c\left(\tau^{i}-\tau_{j}\right)+I_{1}+T+d_{P_{1}}^{i}+d_{j,P_{1}}+\epsilon_{P_{1}}\\ P_{2}=\rho +c\left(\tau^{i}-\tau_{j}\right)+I_{2}+T+d_{P_{2}}^{i}+d_{j,P_{2}}+\epsilon_{P_{2}}\\ \Phi_{1}=\rho +c\left(\tau^{i}-\tau_{j}\right)-I_{1}+T+\lambda_{1}N_{1}+b_{\Phi_{1}}^{i}+b_{j,\Phi_{1}}+\epsilon_{\Phi_{1}}\\ \Phi_{2}=\rho +c\left(\tau^{i}-\tau_{j}\right)-I_{2}+T+\lambda_{2}N_{2}+b_{\Phi_{2}}^{i}+b_{j,\Phi_{2}}+\epsilon_{\Phi_{2}} \end{array}\right.$$
where, Φ denotes the carrier phase observations and 1 and 2 represent the signal frequencies. The superscripts *j* and *i* represent the suitable number of receivers and satellites. *N* and λ are the carrier phase integer ambiguity and the carrier wavelength of GPS L1 or L2. $b^{i}$ and $b_{j}$ are the satellite and receiver fractional ambiguities, respectively. The ionosphere delays can be obtained with dual-frequency ($f_{L1}=1575.42$ MHz, $f_{L2}=1227.60$ MHz) observations and geometry-free (GF) combinations [32] as the following equations:(4)$$P_{4}=P_{1}-P_{2}=\left(I_{1}-I_{2}\right)+\left(d_{P_{1}}^{i}-d_{P_{2}}^{i}\right)+\left(d_{j,P_{1}}-d_{j,P_{2}}\right)+\left(\epsilon_{P_{1}}-\epsilon_{P_{2}}\right)$$
(5)$$\begin{array}{cc} L_{4} & =\Phi_{1}-\Phi_{2}\\ \; & =\left(I_{2}-I_{1}\right)+\left(\lambda_{1}N_{1}-\lambda_{2}N_{2}\right)+\left(b_{\Phi_{1}}^{i}-b_{\Phi_{2}}^{i}\right)+\left(b_{j,\Phi_{1}}-b_{j,\Phi_{2}}\right)+\left(\epsilon_{\Phi_{1}}-\epsilon_{\Phi_{2}}\right) \end{array}$$

Define:

$cDCB^{i}=d_{P_{1}}^{i}-d_{P_{2}}^{i}$, which is the satellite DCB (the unit is the time seconds);

$cDCB_{j}=d_{j,P_{1}}-d_{j,P_{2}}$, which is the receiver DCB (the unit is the time seconds);

$\Delta \epsilon_{P}=\epsilon_{P_{1}}-\epsilon_{P_{2}}$, which are the difference of multipath and noise.

Liu et al. [33] introduced a method that used the carrier phase to smooth the pseudorange. The carrier phase smoothed pseudorange $P_{4,sm}$ observations are shown as follows:(6)$$P_{4,sm}=\left\{\begin{array}{cc} \omega_{t}P_{4}\left(t\right)+P_{4,prd}\left(t\right)\left(1-\omega_{t}\right)\; & \left(t>1\right)\\ P_{4}\; & \left(t=1\right) \end{array}\right.$$
where $\omega_{t}$ is the weight factor involved in epoch *t*th, *t* is the epoch number of the satellite, and $P_{4,prd}$ is the carrier phase prediction pseudorange correction parameter [2,29]:(7)$$P_{4,prd}\left(t\right)=\left\{\begin{array}{cc} \left[L_{4}\left(t\right)-L_{4}\left(t-1\right)\right]+P_{4,sm}\left(t-1\right)\; & \left(t>1\right)\\ 0\; & \left(t=1\right) \end{array}\right.$$

The gross errors and cycle slips are needed to be removed through the GNSS pseudorange (dual-frequency) and ionospheric residuals observations to get the consecutive dual-frequency carrier arc which are used to smooth the pseudorange observations. When *t* is equal to 1, which means the first epoch of one observation arc, $P_{4,sm}$ is equal to $P_{4}$. As the higher orders of ionospheric refraction are very small, in GPS processing, only the first order of ionospheric refraction is used to compute the atmospheric ionosphere delay. The retained ionosphere delay (first-order) can be shown as [34]:(8)$$d_{ion}=40.3STEC/f^{2}$$
where *STEC* is the slant *TEC* and *f* is the frequency of the L1/L2 carrier. Substituting Equation (8) into Equation (7), and replacing $P_{4}$ with the smoothed ionospheric *TEC* and *DCB* measurements $p_{4,sm}$ after processing to ensure that the “carrier phases are aligned to code” (in m), we obtain:(9)$$P_{4,sm}=\left\{\begin{array}{cc} 40.3STEC\left(1/f_{1}^{2}-1/f_{2}^{2}\right)+cDCB_{L_{1}l_{2}}^{i}+cDCB_{j,L_{1}l_{2}}+\Delta \epsilon_{P}\; & \left(t>1\right)\\ P_{4}\; & \left(t=0\right) \end{array}\right.$$

*STEC* can be calculated from dual-frequency GPS observations by Equation (9), and *STEC* can be written as follows:(10)$$STEC=-\left(f_{1}^{2}f_{2}^{2}/40.3\left(f_{1}^{2}-f_{2}^{2}\right)\right)\left(P_{4,sm}-cDCB_{L_{1}l_{2}}^{i}-cDCB_{j,L_{1}l_{2}}\right)$$

The ionospheric layer ranges in altitude from 60∼1000 km. Assuming that there is a concentrated thin shell at altitude H for all electrons in the ionosphere, *STEC* can be translated into the vertical *TEC* (*VTEC*) by the modified single-layer model (MSLM: http://aiuws.unibe.ch/spec/ion.php#processing_description), which is the same as the widely used *STEC* translation model used by the IGS CODE group [31], and is expressed as:(11)$$\left\{\begin{array}{c} STEC=VTEC\cdot MF(z)\\ MF\left(z\right)={\left[1-\sin^{2}\left(\propto z\right)/{\left(1+H_{ion}/R_{E}\right)}^{2}\right]}^{-1/2} \end{array}\right.$$
where, *z* is the elevation angle of visible satellites; $H_{ion}$ is the attitude of the ionosphere thin shell; $R_{E}$ is the Earth’s radius, $R_{E}=6371$ km; ∝ and *H* can be set by users. We define ∝ = 0.9782; $H_{ion}$ = 506.7 km as the values used by the CODE group. The *VTEC*, E(β,s) can be calculated by Equation (12):(12)$$VTEC\left(\beta ,\gamma \right)=\sum_{m=0}^{m_{max}}\sum_{n=0}^{m}{\tilde{P}}_{mn}\left(\sin\beta \right)\left(a_{mn}\cos\left(m\gamma \right)+b_{mn}\sin\left(m\gamma \right)\right)$$
where, γ is the sun-fixed longitude of the ionosphere pierce point (IPP), β is the geocentric latitude of the IPP, and $\gamma =\epsilon -\epsilon_{0}$; ϵ is the longitude of the IPP; $\epsilon_{0}$ is the apparent solar time; $a_{mn}$ and $b_{mn}$ are the regional or global ionosphere model coefficients, respectively. ${\tilde{P}}_{mn}$ is the normalized legendre polynomials:(13)$${\tilde{P}}_{mn}=\Theta \left(m,n\right)P_{mn}$$
where, $P_{mn}$ denotes the un-normalized Legendre polynomials and Θ represents the normalization function:(14)$$\Theta \left(m,n\right)={\left(2\left(2n+1\left(n-m\right)!\right)/\left(1+\delta_{0n}\left(n+m\right)!\right)\right)}^{1/2}$$
with δ being the Kronecker Delta. Substituting Equations (13) and (14) into Equation (12), the following expression can be obtained: (15)$$\begin{array}{cc} \; & -f_{1}^{2}f_{2}^{2}/40.3\left(f_{1}^{2}-f_{2}^{2}\right)\left(P_{4,sm}--cDCB_{L_{1}l_{2}}^{i}-+cDCB_{j,L_{1}l_{2}}\right)\\ \; & =\sum_{m=0}^{m_{max}}\sum_{n=0}^{m}{\tilde{P}}_{mn}\left(\sin\beta \right)\left(a_{mn}\cos\left(m\gamma \right)+b_{mn}\sin\left(m\gamma \right)\right){\left[1-\sin^{2}\left(\propto z\right)/{\left(1+H_{ion}/R_{E}\right)}^{2}\right]}^{-1/2} \end{array}$$
where $P_{4,sm}$ are the smoothed observations, $DCB_{L_{1}l_{2}}^{i}$, $DCB_{j,L_{1}l_{2}}$, $a_{mn}$ and $b_{mn}$ are the unknown parameters needed for estimation. The spherical harmonics expansion (SHE) order depends on the areas of use. Here, for the small areas test, the SHE defaults as a fourth-order. The other orders used are the 8th and 15th orders, depending on the regional, continental, and global scales. More than 20,000 measurements are made every day at one IGS station [2]. The number of observations is enough to estimate the unknown parameters. For a short period of 15 min, the observations are also sufficient to estimate the satellite DCBs. The inequality constraint method has been used here to avoid negative TEC estimates [35]. To analyze the satellite DCBs based on observations of different correlator spacings, the receiver DCBs, satellite DCBs and ionosphere parameters calculated from the GNSS dual-frequency observations by the least-squares (LS) method are discussed in the next section.

### 2.3. Satellite DCB Estimation by a Correlator Spacing Flexible Receiver

For a static station, one satellite can generally be tracked by the station more than once a day since the satellite period is 11 h 58 min. The satellite *DCB* and receiver *DCB* cannot be separated inside a single receiver. The receiver *DCB* is attached to each visible satellite *DCB* (*RASDCB*), and *RASDCB* can be defined as $RASDCB_{j}^{i}=DCB^{i}+DCB_{j}$ for receiver *j* of visible satellite *i*. Assuming *n* RASDCBs need to be estimated from all the stations and setting the *n* rows of *RASDCB* ($Z_{RASDCB}$) as the least squares (LS) adjustment pseudo observations, *RASDCB* can be modeled as:(16)$$\left\{\begin{array}{c} Z_{RASDCB}+V=F\cdot {\hat{X}}_{DCB}\\ {\hat{X}}_{DCB}={\left[{\hat{X}}_{DCB_{sat}}{\hat{X}}_{DCB_{rec}}\right]}^{T} \end{array}\right.$$
where, *V* is the vector of the residuals for pseudo observations and $Z_{RASDCB}$, ${\hat{X}}_{DCB}$ is the vector of estimated DCBs, which contain two parts, the individual satellite *DCB*, ${\hat{X}}_{DCB_{sat}}$, and the individual receiver *DCB*, ${\hat{X}}_{DCB_{rec}}$. *F* is the design matrix, and it is rank-deficient on the order of one and consists of the matrices $F_{sat}$ and $F_{rec}$. $F_{sat}$ and $F_{rec}$ are one for the corresponding matching satellite or receiver and zero for others, so there is only a single element in each row of *F*. Assume the corresponding satellite and receiver numbers are $\mu_{sat}$ and $\mu_{rec}$, respectively. Equation (16) can be changed to:(17)$$\left\{\begin{array}{c} \underset{n\times 1}{\underset{︸}{Z_{RASDCB}}}+V=\underset{n\times (\mu_{sat}+\mu_{rec})}{\underset{︸}{F}}\cdot \underset{(\mu_{sat}+\mu_{rec})\times 1}{\underset{︸}{{\hat{X}}_{DCB}}}\\ \underset{(\mu_{sat}+\mu_{rec})\times 1}{\underset{︸}{{\hat{X}}_{DCB}}}={\left[\underset{1\times \mu_{sat}}{\underset{︸}{{\hat{X}}_{DCB_{sat}}}}\underset{1\times \mu_{rec}}{\underset{︸}{{\hat{X}}_{DCB_{sat}}}}\right]}^{T}\\ \underset{n\times (\mu_{sat}+\mu_{rec})}{\underset{︸}{F}}=\left[\underset{n\times \mu_{sat}}{\underset{︸}{F_{sat}}}\underset{n\times \mu_{rec}}{\underset{︸}{F_{rec}}}\right] \end{array}\right.$$

Therefore, DCBs cannot be estimated through Equation (17) if none of the receiver or satellite DCBs are given as a priori reference values. Generally, three methods are used to eliminate the rank deficiency of Equation (17) [11]: a bias priori fixed receiver;the imposition of a satellite DCBs zero-mean condition;zero references selection from DCB relatively stable satellites.

The GNSS control segments use the first approach to estimate DCBs since they are equipped with precisely calibrated reference receivers. The IGS uses the second approach to separate the receiver and satellite DCBs. More attention must be paid when choosing the zero-mean satellite reference since not all satellites have the same *DCB* stability. The disparate levels of stability of each visible satellite *DCB* will impact the final *DCB* estimation. The influence of satellites with poor stability can be reduced through the selection of relatively stable *DCB* satellites as references [11,30]. The third approach requires a priori analysis of satellite DCBs, and it has not been widely used until now. In this research, we apply the zero-mean satellite reference, the same approach as used by IGS, to separate the receiver and satellite DCBs, as shown in Equation (18).
(18)$$\left\{\begin{array}{c} H\cdot {\hat{X}}_{DCB}=0\\ \underset{(\mu_{sat}+\mu_{rec})\times 1}{\underset{︸}{{\hat{X}}_{DCB}}}={\left[\underset{1\times \mu_{sat}}{\underset{︸}{{\hat{X}}_{DCB_{sat}}}}\underset{1\times \mu_{rec}}{\underset{︸}{{\hat{X}}_{DCB_{sat}}}}\right]}^{T}\\ \underset{1\times (\mu_{sat}+\mu_{rec})}{\underset{︸}{F}}=\left[\underset{1\times \mu_{sat}}{\underset{︸}{e}}\underset{1\times \mu_{rec}}{\underset{︸}{0}}\right],e=\left[\underset{1\times \mu_{sat}}{\underset{︸}{1\cdots 1}}\right] \end{array}\right.$$
where *H* is the constraint vector, and the elements corresponding to ${\hat{X}}_{DCB_{sat}}$ are 1 and 0 for others.

For a position-fixed single receiver or station, no more than 12 GPS satellites are visible at the same time. The constraint condition discussed above is not fit under this situation. Here, the IONEX files from the previous day are imported to validate our estimation under the above constraint conditions [36]. A short period of approximately 15 min can reflect the characteristics of satellite DCBs since we assume that the satellite DCBs remain static during a day, the sum of all GPS satellite DCB values is zero, and the satellite DCBs are static during 24 h [37]. The DCBs of the invisible satellites are set as known parameters using the IONEX file. The MSDCB method is based on the pseudorange observations calculated by the method discussed in the section “Correlator spacing influence on pseudorange measurements”. A flowchart of the single receiver multi-spacing DCB (MSDCB) estimation is shown in Figure 2 [2]. The MSDCB method contains two parts, (1) data collection and processing; (2) DCB estimation introduced in the section “GNSS observation and pre-processing” and “Satellite DCB estimation by a correlator spacing flexible receiver [22,23,38,39,40]”, and consists of the following steps:

data collection and processing, collect IONEX files, MGEX sp3 files, and receiver coordinates;collect IF data with dual-frequency front-end;using a spacing flexible software receiver to get observations of different spacing;local ionospheric TEC modeding;DCB estimation by Least Squate estimation method.

**Figure 2 sensors-21-00443-f002:**
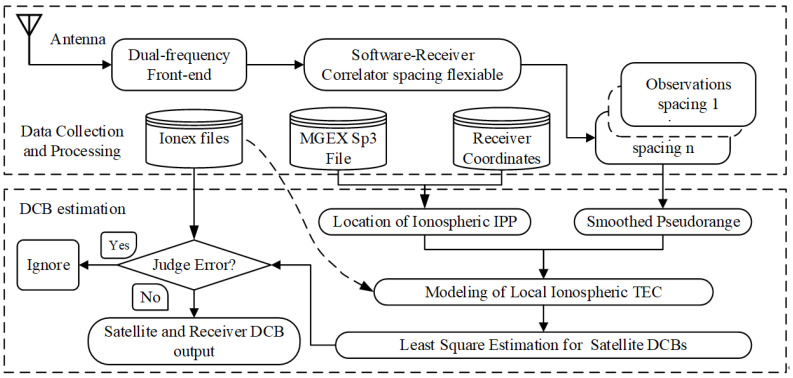
Flowchart of single receiver multi-spacing Differential Code Biases (DCB) estimation.

## 3. Experiment, Result Comparison, and Analysis

In this section, first, the IF data collection experiment is introduced. Then, the characteristics of the satellite DCBs estimated from 24 h and 15 min are analyzed. Next, the multi-station DCBs estimated by four IGS stations and correlator spacing changeable dual-frequency software receiver observations are assessed. Finally, the DCBs estimated by the IGS stations and changeable software observations are calculated and compared with existing products.

### 3.1. Experimental Outline

To estimate and analyze the satellite and receiver DCBs by a correlator flexible single receiver, a dual-frequency front-end (Stereo V2) and a software receiver are used to collect and process the IF data. Satellite DCBs are estimated by the multi-station method and single-station method using data downloaded from the IGS reference stations. The DCBs estimated by multi-station, single-station, and MSDCB are evaluated using IGS Analysis Center products.

A software receiver is used to process the dual-frequency IF data, and the output includes observations of the GPS L1 C/A and L2 CM/CL signals. GPS RINEX observations containing C1 and C2 observations are used in this experiment, but the C2 observations are replaced by P2 observations when C2 observations are not available [41]. We collected the dual-frequency GPS IF data from the roof of the Engineering Building at the University of Calgary. The antenna coordinates are 1,641,890.08105434 m, −3,664,879.34459194 m, and 4,939,969.42850734 m under the WGS84 coordinate system. The IF data sampling frequency was 26 MHz; the data set was as large as 800 MB per minute. Several 30 min long (30 GB) datasets were collected with the dual-frequency front-end. The left image in Figure 3 show the Skyplot of visible satellites from the IF data from 21:19 to 21:34, 16 August 2017. During these 15 min, ten satellites were observed by the receiver. In this experiment, the cutoff elevation angle was chosen to be 20 degrees to mitigate the impact of multipath noise. After cycle slip removal and smoothing, only observations from five satellites were useful, as shown in the right image in Figure 3 [42]. Since the observations to satellite PRN 31 were very short and therefore were ignored in this research, the characteristics of the other four satellite DCBs were analyzed.

Five IGS stations (http://www.igs.org/network WILL, SASK, PRDS, DRAO, BREW) around Calgary are selected to estimate the multi-station satellite DCBs, and these stations are shown in Figure 4. The RINEX files (ftp://cddis.gsfc.nasa.gov/pub/gps/data/daily/), the IONEX data (ftp://cddis.nasa.gov/pub/gps/products/ionex/), and the MGEX clock product Standard Product 3 (SP3) files (ftp://cddis.gsfc.nasa.gov/pub/gps/products/week/) are used in our experiment.

### 3.2. Stability of 15 Min Estimated DCBs

To study the characteristics of the satellite DCBs estimated by 15 min data, DCBs estimated from multiple stations and a single station are employed and analyzed using 24 h and 15 min data (16 August 2017). The DCBs from the CODE product are selected as the correct reference. The 24 h data analysis aims to prove that our software is useful and accurate enough to estimate satellite DCBs. The DCBs estimated from 15 min data are reliable and are compared in the next section. Since satellite DCBs are assumed to be stable for one day, the 15 min data estimated DCBs can reflect the characteristics of DCBs and can improve the real-time performance of DCBs if they can be estimated using only 15 min GNSS observations. The results are compared with the CODE products. The DCB values estimated from 5 IGS stations (WILL, SASK, PRDS, DRAO, BREW) and a single station (PRDS) and those released and broadcasted by CODE products are shown in Figure 5 (24 h) and Figure 6 (15 min).

The root mean square (RMS) and mean difference of these differences between multi-station estimated DCBs (24 h) and those released from CODE are shown in the right two images in Figure 5. The RMS values of all 32 satellites are less than 0.3 ns, and 17 have an RMS less than 0.1 ns; only the RMS values of PRN 9 and PRN 26 are larger than 0.2 ns. The absolute values of the mean differences of PRN 9 and PRN 26 are larger than 0.2 ns, and those of the other satellites are all less than 0.2 ns. The RMS of the single-station estimated DCBs and those released from CODE are less than 0.3 ns, and only the RMS of PRN 26 is larger than 0.2 ns. The absolute values of the single-station estimated DCBs are less than 0.2 ns except PRN 26 (larger than 0.2 ns), as shown in the left two images in Figure 5. In summary, a single station can obtain reliable results. These test results show good agreement with the CODE products for both multi-station and single station estimated DCBs, which indicates that the algorithm and software are accurate enough to analyze the characteristics of short time (15 min) estimated DCBs. In addition, the performances of some of the satellite DCBs estimated from single-station data are better than those estimated from multiple stations (5 stations), which seems counterintuitive, as more stations mean more and better distributed IPPs. The reason for this result may be that the region of the 5 stations of multi-station DCB estimation is not wide enough. More attention will be paid in further research to judge how many stations are suitable to estimate satellite DCBs.

The RMS and mean difference of these differences between multi-station 15 min data (top two graphics) and single-station 15 min data (bot two graphics) estimated DCBs and those released from CODE are shown in Figure 6. The DCBs of seven satellites can be estimated with 15 min data. For the multi-station estimated DCBs, the RMS values of the four satellites are less than 1.0 ns, and those of the other three satellites are less than 3.0 ns. The absolute value of the mean difference is <4 ns, and four of them have an absolute mean difference of less than 1.0 ns. For the single-station estimated DCBs, the RMS values of six satellites are less than 1.0 ns, only the RMS of PRN 15 is larger than 2.0 ns, and three of them have an RMS value of less than 0.5 ns. The absolute value of the mean difference has the same characteristics as the RMS. Previous results have demonstrated that satellite DCBs (15 min) estimated from a single station are better than those estimated using the multi-station method. Visible satellites of a single station may have a better geometric distribution than those of multiple stations since they are in the middle of the five stations. The RMS and mean difference of the 15 min estimated satellite (observations continued satellites in the right image in Figure 3) DCBs are listed in Table 1. The DCBs estimated from 15 min data can be a reflection of the DCB characteristics and are compared with the DCBs estimated by the MSDCB method.

### 3.3. Correlator Spacing Influence on the Multi-Station Estimated DCBs

To analyze the effect of the correlator spacing on multi-station DCB estimation, four IGS stations (WILL, SASK, DRAO, BREW) and a correlator spacing flexible receiver (yellow point in Figure 4) are used to estimate multi-station DCBs. The current day CODE DCB products are used as a reference. Five correlator spacings (0.6 to 1.0) are used to calculate GNSS observations since the observations are not accurate enough when the correlator spacing is less than 0.6 chips [21,28]. Observations calculated from the same spacing for all satellites are used as an independent receiver. The RMS and mean difference of the differences between the correlator spacing flexible receiver-based multi-station estimated DCBs and those released from CODE are shown in Figure 7 and Figure 8, respectively. Their performance is at the same level as shown in Figure 6 (top 2 graphics), less than 3.0 ns.

The RMS and mean difference show a significant difference with different spacings. The smallest RMS and mean difference occurred with spacings of 0.8 chips and 1.0 chips (two satellites each), respectively; the RMS and mean difference values were all less than 1.0 ns, and three of the satellites had values were less of than 0.4 ns, only the values of PRN 29 is 1.0 ns. The RMS and mean difference calculated from spacing of 0.8 chips have two best and two worst DCBs, more researches should be done for this spacing. The RMS and mean difference calculated from spacing of 0.7 chips have the two worst and one second-worst values. They are all larger than 1.2 ns. The RMS and mean difference of all the satellites of spacings of 0.6 chips and 0.9 chips are all at the middle level compared with the other spacings. These two spacings can be chosen as the fixed spacing for a structure fixed receiver. For satellites, most of the best DCBs from different spacings are better than those estimated from multi-station data, as shown in the left two columns of Table 1. The reason for this result is likely that some of the correlator spacing-induced pseudorange satellite-dependent biases merge into the IPPs and influence the distribution of IPPs. The observations of each station determine the performance of the multi-station -estimated satellite DCBs. For the multi-station (5 stations) DCB estimation, an observation change of one station can cause DCB variations. The performance can be improved with the best correlator spacing of each station.

### 3.4. Correlator Spacing Flexible Receiver-Based Single-Station Multi-Spacing Estimated DCBs

The main idea of the MSDCB method is to improve the IPP distribution and the performance of the satellite DCBs estimation by increasing the number of observations inside a single receiver. To analyze the performance of the MSDCB method, the MSDCB estimated DCBs are compared with the multi-station estimated DCBs, the single-station estimated DCBs, and the current day CODE products. The RMS and the mean difference of these differences between correlator spacing flexible receiver-based single-station multi-spacing estimated DCBs (MSDCB) and those released from single-station (15 min), multi-station (15 min), and CODE are shown in Figure 9. The RMS between MSDCB and those released from single-station are all less than 1.3 ns two of them are larger than 1.0 ns, and the mean difference (absolute value) between MSDCB and those released from single-station are all larger than 3.0 ns. These results show that the MSDCB is different from those DCBs estimated from single-station short time period (15 min) data, and the MSDCB method is different from the single-station method. The RMS between MSDCB and those released from 15 min multi-station data are all less than 1.6 ns, two of them are less than 0.5 ns, and the mean difference (absolute value) between MSDCB and those released from multi-station are all less than 2.4 ns and two of them are less than 0.7 ns. The previous results have demonstrated that the satellite DCBs calculated by the MSDCB method are closed to the satellite DCBs obtained from the multi-station method. The MSDCB method improves the IPP distribution of a single-station.

The RMS between MSDCB and those released from CODE products are all less than 2.7 ns, two of them are less than 0.7 ns, and the mean difference (absolute value) between MSDCB and those released from multi-station are all less than 2.4 ns and three of them are larger than 1.8 ns. The MSDCBs of PRN 5 and PRN 21 are accurate enough and could be used for resolution and they have nearly the same performance compared with those shown in Table 1. The RMS and mean difference (yellow bar in Figure 9) of PRN 26 and 29 are more like those of the multi-station method estimated DCBs (Columns 2 and 3 in Table 1) than the single-station method estimated DCBs (Columns 4 and 5 in Table 1). The comparison of the RMS and mean difference values between DCBs calculated by the MSDCB and those released for the CODE products (given in Table 1) has demonstrated that the MSDCB method results are similar to the multi-station method. The MSDCB method can be used to estimate satellite DCBs (15 min) since the DCBs estimated by the MSDCB method have the same performance level as the multi-station method and the single-station method (Table 1). The MSDCB method augments the observation quantity for every visible satellite than the single-station method. It could offer better IPPs distribution than the single-station method. The main novelty of the MSDCB method is to increase the observation quantity of one receiver to improve the performance of the estimated satellite DCBs. However, the performance of the MSDCB method is not accurate enough. The result of MSDCB should be better if it increases the epoch number. There are only DBCs of four satellites that have been estimated for the 15 min data; the number of satellite DCBs estimated by the MSBCD method could be improved if the conditions of measurement pre-processing are released.

## 4. Conclusions

In this contribution, a new method is proposed for estimating and analyzing satellite DCBs. The new method is based on GNSS observations from a correlator spacing flexible software receiver instead of GNSS observations conventionally from a network of hardware receivers. Since receiver measurements vary with correlator spacing [21], it is theoretically possible to analyze the effect of correlator spacing on satellite DCB estimation and future estimate satellite DCBs using observations from a correlator spacing flexible software receiver. In the analysis of the influence of the correlator spacing on multi-station estimated DCBs by changing the spacing of a receiver in the multi-station DCB estimation process, the results show a spacing-dependent difference. The RMS values between the best and the worst estimated DCBs are larger than 1.0 ns for all the satellites, and the most accurate satellite DCBs do not all occur with the same spacing compared to the CODE products. The satellite DCBs estimated by the MSDCB method have the same performance as the DCBs estimated by the multi-station and single-station (correlator spacing fixed receiver) approaches after comparison to the CODE products.

Based on the analysis concerning the CODE DCB products, some conclusions can be drawn. The precision of satellite DCB estimates obtained by multi-station and single-station short time data can reach around 1.0 ns. The correlator spacing changing of a receiver in the multi-station (5 stations) DCB estimation approach estimated satellite DCBs shows a significant spacing difference. Inconsistent correlator spacing will lead to inconsistencies in the DCB estimation for a network with mixed receiver spacing. For the future, the research on the classification of correlator spacing into consistent groups, and the estimation of DCBs for each spacing group are necessary for multi-station DCB estimation should be settled. A comparison of the DCB estimates using the proposed MSDCB method to the DCB estimates from the multi-station method and the single-station method demonstrates that the MSDCB method can promote the single-station method IPP distribution close to the multi-station method IPP distribution. The MSDCB method can reach the same performance as the multi-station DCB estimation method. It is a new approach that use the observations measured by different correlator spacings with only one software receiver. The real-time performance of satellite DCB estimation can be improved if the MSDCB method can be implemented into hardware receivers. It is also necessary to note that the MSDCB method (for 15 min data) is still not accurate enough, and more works are needed to improve it further. The receiver DCB estimation using the MSDCB method should also be considered in the future.

## Figures and Tables

**Figure 1 sensors-21-00443-f001:**
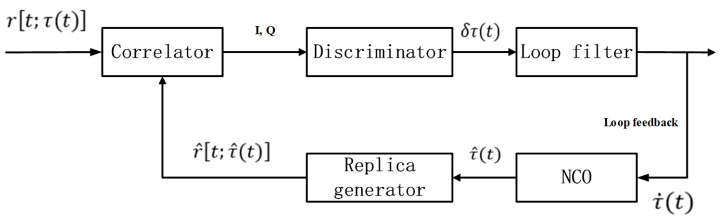
Linear model of a delay locked loop (DLL)-based code tracking loop under the time domain.

**Figure 3 sensors-21-00443-f003:**
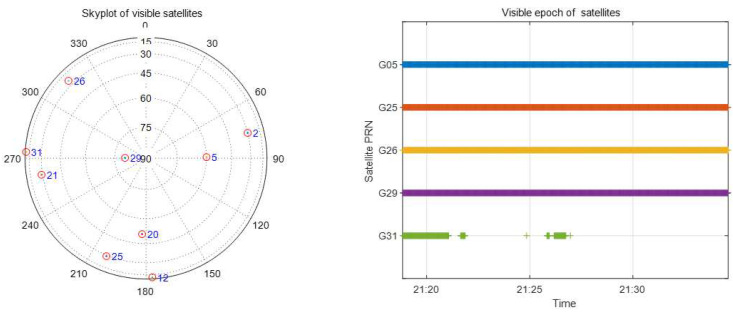
Skyplot and epoch of visible satellites 21:19–21:34, 16 August 2017.

**Figure 4 sensors-21-00443-f004:**
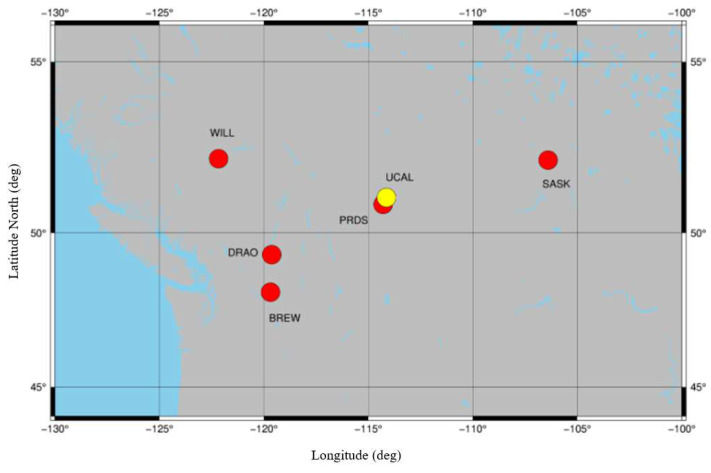
Distributions of the Chosen International GNSS Service (IGS) Stations.

**Figure 5 sensors-21-00443-f005:**
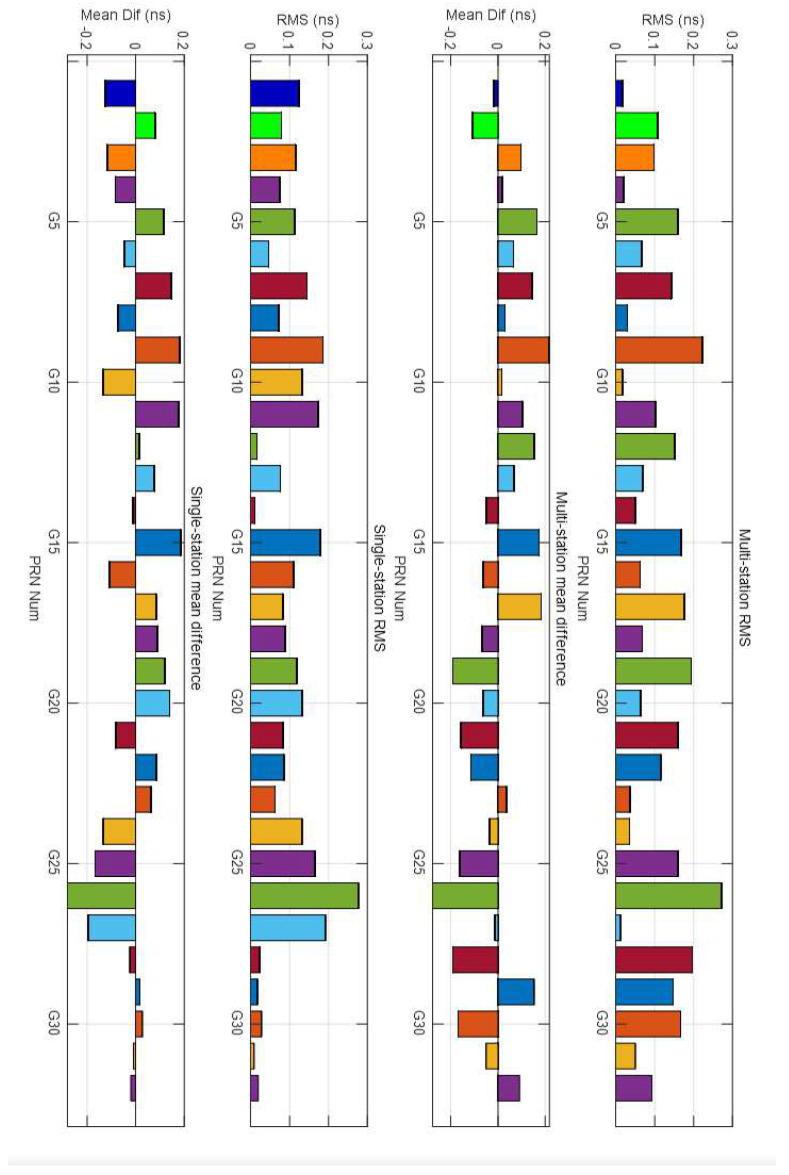
Root mean square (RMS) and mean difference between the multi-station estimated satellite DCBs and Center for Orbit Determination in Europe (CODE) products (**right** 2 plot), and RMS and the mean difference between the single-station estimated satellite DCBs and CODE products (**left** 2 plot) from 16 August 2017.

**Figure 6 sensors-21-00443-f006:**
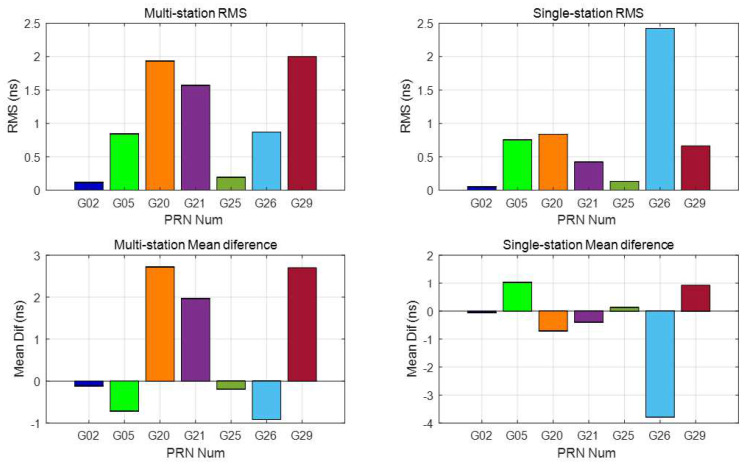
RMS and the mean difference between multi-station (**left** 2 plot) and single-station (**right** 2 plot) short period (15 min) estimated satellite DCBs and CODE products for 21:19–21:34, 16 August 2017.

**Figure 7 sensors-21-00443-f007:**
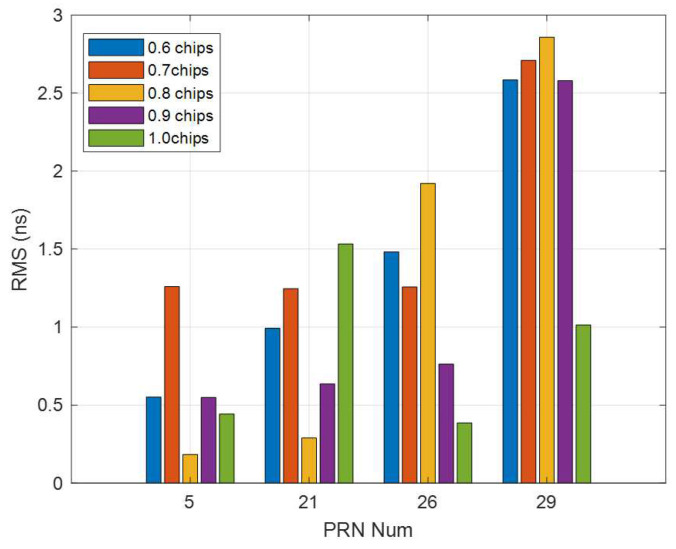
RMS between multi-station (4 IGS stations and 1 correlator spacing flexible receiver) estimated satellite DCBs and CODE products for 21:19–21:34, 16 August 2017.

**Figure 8 sensors-21-00443-f008:**
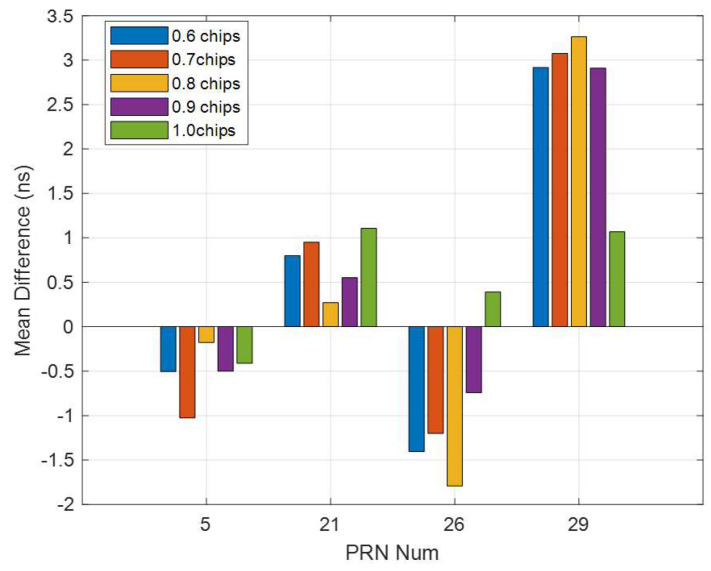
Mean difference between multi-station (4 IGS stations and 1 correlator spacing flexible receiver) estimated satellite DCBs and CODE products for 21:19–21:34, 16 August 2017.

**Figure 9 sensors-21-00443-f009:**
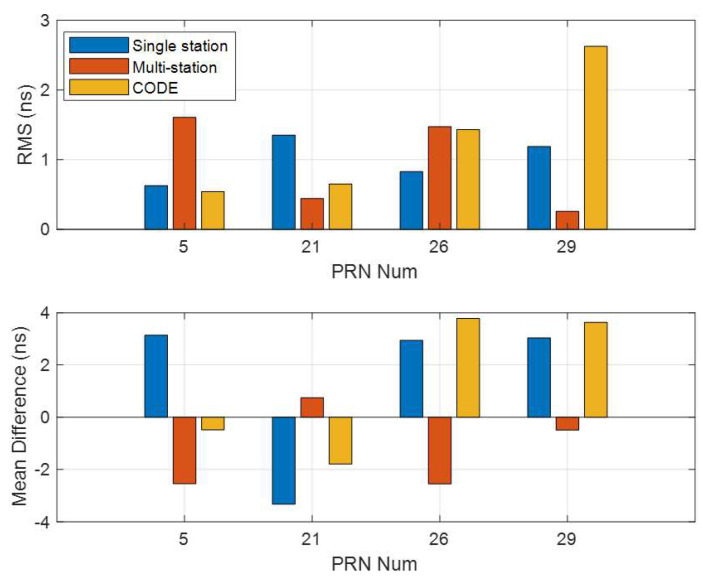
RMS (**top** plot) and mean difference (**bottom** plot) between correlator spacing flexible receiver estimated satellite DCBs and multi-station estimated DCBs, single-station estimated DCBs, and CODE products for 21:19–21:34, 16 August 2017.

**Table 1 sensors-21-00443-t001:** Satellite DCB RMS and mean differences between satellite DCB estimates from 16 August 2017, using the multi-station method, single-station method and CODE for 15 min data.

PRN	M-RMS (ns)	M-Mean (ns)	S-RMS (ns)	S-Mean (ns)
05	0.8426	−0.3588	0.7519	0.5125
25	0.1916	−0.0970	0.4204	−0.2970
26	0.8696	−0.4573	2.4240	−1.8955
29	2.9393	2.1005	0.6646	0.4640

## Data Availability

Publicly available datasets were analyzed in this study. This data can be found here: https://www.igs.org.

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
