# Peer review of "Estimation and Analysis of GNSS Differential Code Biases (DCBs) Using a Multi-Spacing Software Receiver"

_sensors, 2021, doi:10.3390/s21020443_

Round 1

Reviewer 1 Report

The current paper proposes a new method is proposed for estimating and analyzing satellite DCBs. The algorithm is validated through experimental results.

Comments to authors:

- The authors can add the steps of implementing the algorithms. The theoretical part can be better detailed. The steps will be in the benefit of the readers, maybe they’ll help the readers to implement the proposed algorithm.

- Please add more details of how the theory from the first sections is applied in the results section.

- Please add the units of measurement both abscissa and orderly in all figures.

- Please add more details regarding the obtained results.

- The state of the art it is very poor, maybe the author could add the following publications:

o Hybrid Data-Driven Fuzzy Active Disturbance Rejection Control for Tower Crane Systems, European Journal of Control, doi https://doi.org/10.1016/j.ejcon.2020.08.001, pp. 1-11, 2020.

o Event-Triggered Adaptive Fuzzy Control for Stochastic Nonlinear Systems with Unmeasured States and Unknown Backlash-Like Hysteresis, IEEE Transactions on Fuzzy Systems, doi 10.1109/TFUZZ.2020.2973950, pp. 1–19, 2020.

- Please add more details regarding the obtained results.

- The authors could add a paragraph with the disadvantages of the proposed method. In the proposed manuscript only the advantages are presented.

- Please add more details regarding paper’s novelty.

Author Response

Dear Reviewer:

The authors would like to thank the Editor and the Reviewers for reviewing the
manuscript and for providing valuable suggestions on how to improve its quality. Those comments are valuable and have helped us further revise and improve this paper.

Reviewer 2 Report

The paper presents a new method is proposed for estimating and analysing satellite differential code biases. The topic is relevant for the journal Sensors and the paper is presented in a professional style. The review of the literature is adequate and the charts are clear. The reviewer has merely some minor recommendations:

  • The extensive use of acronyms, namely in title and in abstract from 1st line and without previous definition lead to a poor readability.In what concerns the title the use of the acronym "GNSS" may be not the best practice. The same applies to the definition of the acronym DCBs in the title (not usual). In teh abstract we have in 1st line the acronyms TEC, DCB and GNSS, just as an example. Limit the acronyms to the minim necessary and define them in abstract and in text (again).
  • The grammar is in general acceptable. However, long phrases should be avoided.
  • Usually units are not in italic
  • All math symbols, for variables and operators, should be clearly defined in the text. For example the large black circle in Eq (11) apparently stands for multiplication. A conventional small dot (in Latex \cdot) or even not Dort at all seem preferable, otherwise readers are mislead and expect some other mathematical operation
  • Sine the paper seems written in Latex/Bibtex the reviewer recommends some tricks, such as, for example,  \sin and \cos (instead of simply sin and cos) or in references use {} to preserve caps [5] -> Radio science -> Radio {S}cience, or ref [32]->Gps Solutions->{GPS} Solutions
  • All plots should have labels (and eventually units) in axes. For example in Fig 4
  •  Some typos should be checked. For example, Eq (12) ( m\gamma), delete unnecessary space, or line 255 missing space (apparently)
  • Check equations that need parenthesis. For example line 159, Eq. 9 should be Eq. (9).

Author Response

(The authors gave the same response as above.)

Round 2

Reviewer 1 Report

In this revision the paper has been seriously improved. The authors answered to all my concerns and from my point of view the paper can be published in Sensors journal.